# Characterization of Chitin Synthase B Gene (*HvChsb*) and the Effects on Feeding Behavior in *Heortia vitessoides* Moore

**DOI:** 10.3390/insects14070608

**Published:** 2023-07-05

**Authors:** Qingling Chen, Mingxu Sun, Hanyang Wang, Xiaohan Liang, Mingliang Yin, Tong Lin

**Affiliations:** College of Forestry and Landscape Architecture, South China Agricultural University, Guangzhou 510642, China; chenqingling@stu.scau.edu.cn (Q.C.); isunmingxv@icloud.com (M.S.); wanghanyang@stu.scau.edu.cn (H.W.); 2020liangxiaohan@stu.scau.edu.cn (X.L.); mingliang.yin@scau.edu.cn (M.Y.)

**Keywords:** *Heortia vitessoides* moore, chitin synthase B gene, RNA interference, starvation, re-feeding

## Abstract

**Simple Summary:**

*Heortia vitessoides* Moore is a leaf-eating pest that affects *Aquilaria sinensis*. In the outbreak period, the leaves of *Aquilaria sinensis* can be eaten up in a short time, resulting in the death of trees and great economic losses. Chitin is the main component of insect cuticle, peritrophic membrane, and tracheal intima. Chitin synthesis in insects is a complex process that requires the cooperation of many enzymes. Chitin synthase is one of the key enzymes in the process. Chitin synthase is divided into two types: chitin synthase A gene (*Chsa*) and chitin synthase B gene (*Chsb*). *Chsb* is mainly responsible for the tissue synthesis of chitin in midgut peritrophic membrane. It was found that the expression of *HvChsb* was inhibited, the growth and development were abnormal, and the mortality rate was increased. These findings provide a reference for the prevention and control of the pest from the perspective of gene manipulation.

**Abstract:**

The chitin synthase B gene is a key enzyme in the chitin synthesis of insect peritrophic matrix (PM), which affects insects’ feeding behavior. The chitin synthase B gene was cloned from the transcription library of *Heortia vitessoides* Moore. RT-qPCR showed that *HvChsb* was highly expressed in the larval stage of *H. vitessoides*, especially on the first day of the pre-pupal stage, as well as in the midgut of larvae and the abdomen of adults. After starvation treatment, *HvChsb* was found to be significantly inhibited over time. After 48 h of starvation, the feeding experiment showed that *HvChsb* increased with the prolongation of the re-feeding time. The experimental data showed that feeding affected the expression of *HvChsb*. *HvChsb* was effectively silenced via RNA interference; thus, its function was lost, significantly decreasing the survival rate of *H. vitessoides*. The survival rate from larval-to-pupal stages was only 43.33%, and this rate was accompanied by abnormal phenotypes. It can be seen that *HvChsb* plays a key role in the average growth and development of *H. vitessoides*.

## 1. Introduction

Chitin is the second most abundant organic compound after cellulose, and it is found in various organisms. It is a linear polymer of β-(1,4)-*N*-acetyl-D-glucosamine (GlcNAc) [1]. It is widely found in arthropods, as well as in some invertebrates, fungi, protozoa, and algae [2,3]. In insects, chitin plays essential roles in providing protection, support, and nutrition. For example, it is the main structural component of insect epidermis and peritrophic matrix (PM), being closely related to growth and development [4,5]. Chitin synthesis is a complex process that requires the co-operation and participation of many enzymes, in which chitin synthase (CHS) is indispensable [1,6]. Previous studies identified genes encoding chitin synthase in many insects, with the findings showing that there are two chitin synthase types: chitin synthases A (*Chsa*) and B (*Chsb*), which are now known as *Chs1* and *Chs2* [7,8,9].

Past studies found that these two CHS exhibit significant differences in mRNA expression specificity and function. *Chsa* is expressed explicitly in the formation of trachea and integument, while *Chsb* is mainly expressed in the midgut [10,11,12]. The two genes also differ markedly in terms of their physiological functions: *Chsa* is mainly involved in the tissue synthesis of the epidermis and trachea at various stages of insect growth, while *Chsb* is mainly responsible for synthesizing chitin in the PM upon eating [13,14,15]. Studies on the *Chsb* gene in insects confirmed its functionality [16,17,18]. Breakthroughs were also achieved using RNAi technology to silence *Chsb* function in insects. Analysis of the expression of *Chs2* in *Locusta migratoria* by Xiaojian Liu showed that *LmChs2* expression was not detected in the pre- and mid-egg stages, while the expression level increased sharply during late-egg development and was stable in the L4, L5, and adult stages. RNAi technology was also used to inject ds*LmChs2* into female and male adults. Compared to the level in the control group, it was found that the expression of this gene was significantly reduced, while feeding noticeably decreased and the mortality of female and male adults increased [19]. In addition, Arakane et al. performed RNAi on *Chsb* of *Tribolium castaneum*. This experiment led to an absence of PM formation in the midgut of larvae of *T. castaneum*, which resulted in reduced larval growth due to starvation. *Chsb*, thus, greatly impacts insect feeding behavior and plays an important role in insect growth and development [20]. The study of this gene, thus, has major biological significance. To date, *Chsb* has been partially characterized in *Locusta migratoria, Spodoptera exigua, Ostrinia furnacalis, Bombyx mori*, and other insects; however, it has not been reported in *H. vitessoides* [15,18,19,21].

*Aquilaria sinensis* (Lour.) Spreng. (Myrtales: Thymelaeaceae: Aquilaria) is an economically important evergreen tree [22,23]. *Heortia vitessoides* Moore (Lepidoptera: Crambidae: Odontiinae) is a serious leaf-eating pest characterized by eating large amounts of leaves in a short period, causing damage that lasts for a long period of time. This moth uses *A. sinensis* as its only food source [24]. In this study, RT-qPCR was used to detect the expression specificity of *HvChsb* in *H. vitessoides* in different stages and tissues. After silencing *HvChsb* via RNAi, its role in the growth and development of *H. vitessoides* was analyzed. At the same time, the expression level of *HvChsb* upon starvation treatment was measured, which further confirmed the importance of this gene in insect feeding behavior. This work provides a scientific basis for exploring the prevention and control of *H. vitessoides* via molecular biological methods.

## 2. Materials and Methods

### 2.1. Insects

The insects were kept in a climatic cabinet (27 °C with 70% relative humidity and a photoperiod of 14:10 h light:dark) and fed *A. sinensis* leaves. When the larvae matured, they were transferred into a container that was filled with sand kept at a humidity of 50% and thickness of 2–4 cm, where they were allowed to undergo pupation and eclosion.

### 2.2. Sample Preparation

To study the expression characteristics of the target gene in each developmental stage of *H. vitessoides*, 90 first-instar larvae (3 biological replicates, 30 per replicate); 45 s-instar larvae (3 biological replicates, 15 per replicate); 6 each of third-, fourth-, and fifth-instar larvae; 6 pupae; and 6 adults (three biological replicates, two per replicate) were analyzed. Secondly, to examine the tissue-specific expression of the target gene, the tissues of L5 larvae and adults were dissected and collected. For starvation treatment, 90 of the L4 larvae (three biological replicates, 30 per replicate) were deprived of food for 96 h and sampled every 12 h. For re-feeding after the starvation treatment, 60 of the L4 larvae (three biological replicates, 30 per replicate) were fed after being deprived of food for 48 h and samples were collected after 0.5, 1, 4, and 12 h. These samples were wiped clean with sterile cotton balls, treated with liquid nitrogen, and, finally, stored in a freezer at −80 °C.

### 2.3. Sequence Verification and Phylogenetic Analysis

A search for the gene sequence in the transcriptome of *H. vitessoides* was performed [25]. After BLAST homology alignment via the NCBI website, the complete sequence of the *Chsb* gene was obtained and named *HvChsb* (accession number: ON783456). The cDNA sequence of the *Chsb* open reading frame was also acquired using the Open Reading Frame (ORF) finder tool (http://www.ncbi.nlm.nih.gov/gorf/gort.html (accessed on 2 August 2022)). Corresponding gene-specific primer pairs were designed to amplify the *Chsb* ORF for sequence verification (Table 1). Primer Premier 5.0 (Premier Biosoft International, Palo Alto, CA, USA) software was used to design *HvChsb*-specific primer sequences. PCR amplification conditions were set as follows: 98 °C for 3 min; 15 cycles starting at 98 °C for 20 s, 66 °C for 10 s, and 72 °C for 15 s, with a decrease in temperature of 1 °C per cycle; 25 cycles of 98 °C for 20 s, 51 °C for 10 s, and 72 °C for 15 s; and one cycle at 72 °C for 2 min before being held at 12 °C. The product was recovered, purified, ligated with pClone007, and transferred into *Escherichia coli* DH5α-competent cells. Finally, it was sequenced to confirm that the target gene had been successfully cloned. The isoelectric protein point and relative molecular weight of *HvChsb* were predicted via the ExPASy-ProtParam tool (http://web.expasy.org/protparam/ (accessed on 29 August 2022)), the prediction of the transmembrane domain of gene protein was performed via TMHMM, and the glycosylation site prediction for this gene was performed via the NetNGlyc 1.0 Server. The amino acid sequences encoded by other insect *Chsb* genes were downloaded from the GenBank database for phylogenetic tree construction and homology comparison. The sequences were aligned on the MAFFT version 7 website. The phylogenetic tree was constructed via MEGA 7.0 software (Mega Limited, Auckland, New Zealand) based on the neighbor-joining method.

### 2.4. RNA Extraction and cDNA Synthesis

Total RNA Kit II (OMEGA) was used to extract total RNA from the sample. Next, the concentration of the extracted RNA was determined using the Implen Ultramicro-spectrophotometer (Nanophotometer series). The PrimeScript^TM^ RT Reagent Kit with gDNA Eraser kit was then used to synthesize cDNA following its operating instructions. The synthesized cDNA was stored in a freezer (−20 °C) for later use.

### 2.5. Primer Design and Quantitative Real-Time Polymerase Chain Reaction (RT-qPCR)

Under the conserved region, specific primers were designed using Primer Premier 5.0 software (Premier Company, Canada), the synthesis of which was then outsourced to Guangzhou Qingke Biotechnology Company. The primer sequences are shown in Table 1. The previously synthesized cDNA templates were diluted to create RT-qPCR reaction templates. The instrument LightCycler 480 II Real-Time PCR System was used to conduct quantitative fluorescence analysis. Three technical replicates were established, with β-actin [26] being used as an internal reference gene.

### 2.6. dsRNA Preparation and Injection

The synthesis of dsRNA was performed using the T7 RiboMAX^TM^ Express RNAi System kit. Primers containing the T7 polymerase promoter sequence were synthesized to run PCR to obtain DNA templates, after which ds*Chsb* and ds*GFP* fragments were together synthesized. The DNA template was removed, followed by removal of dsRNA annealing and single-stranded RNA (ssRNA), and, finally, dsRNA was purified. The purified dsRNA was diluted with nuclease-free water and quantified using an Implen Ultramicro-spectrophotometer (Nanophotometer series).

The dsRNA was diluted to a concentration of 3 μg/μL, and 1 μL was injected into the dorsal part of the antepenultimate abdominal segment of each larva using a microinjector. The same concentration and dose of ds*GFP* and DEPC were used in the control group. Each group contained at least 30 larvae, and four boxes of injection were used to record the phenotypic changes and survival rates that occurred during the experiment.

### 2.7. Phenotype Observation and Analysis

Careful observation of the phenotypic changes was performed in the experimental and control groups after injection. To determine whether the treated insects survived, we touched them with a brush to determine whether they responded within 1 min.

### 2.8. Starvation Treatment and Re-Feeding

L4D1 larvae were randomly selected and divided into three groups, with 30 allocated to each group, and deprived of food for 96 h. The samples of starved larvae were collected at 12, 24, 36, 48, 72, and 96 h. The duration of starvation varied, while the other conditions remained unchanged. *H. vitessoides* that were fed during the same period were collected as a control group. The L4D1 larvae were selected and divided into three groups, with 20 larvae allocated to each group. After 48 h of starvation, they were again fed. Samples from the re-fed larvae were collected at 0.5, 1, 4, and 12 h, and the *H. vitessoides* usually reared in the same period was used as a control group. These collected samples were quickly frozen in liquid nitrogen and stored in an ultra-low-temperature freezer (−80 °C).

### 2.9. Statistical Analysis

Excel was used to perform the primary statistical analyses of experimental data, and SPSS 18.0 (IBM, Armonk, NY, USA) was used to perform statistical analyses. Tukey’s test and one-way analysis of variance (ANOVA) were used to conduct analyses among multiple samples, while the *t*-test was used to conduct analyses among two samples. In the software, the 2^−ΔΔ Ct^ data analysis method was used to obtain the relative expression of the target gene [27]. At *p* < 0.05, the difference was statistically significant. The data obtained are expressed as mean ± standard error.

## 3. Results

### 3.1. Sequence Analysis of HvChsb and Phylogenetic Analysis

The gene sequence was searched in the transcriptome of *H. vitessoides*. After BLAST homology alignment via the NCBI website, the complete sequence of the *Chsb* gene was obtained and named *HvChsb* (GenBank accession number: ON783456). The full length of the sequence was 4971 bp, and the sequence had an ORF of 4410 bp, which encoded 1469 amino acids. With the help of the online program ProtParam, the theoretical molecular weight of the protein encoded by the gene was 168.11 kDa, which had a predicted isoelectric point of 5.97. Among the residues, the negatively charged residues (Asp and Glu) numbered 172, while the positively charged residues (Arg and Lys) numbered 154. Analysis via TMHMM showed that *HvChsb* had 16 transmembrane domains (Figure 1).

The amino acid sequences of six insects were downloaded from GenBank (*Helicoverpa armigera*, *Cnaphalocrocis medinalis*, *Mythimna separata*, *Tribolium castaneum*, *Bactrocera dorsalis*, and *Locusta migratoria*). The results showed that the values of amino acid sequence similarity between *HvChsb* and the *Chsb* genes of these insects were 70.64%, 73.37%, 68.31%, 43.06%, 38.22%, and 45.95%, respectively (Figure 2).

To understand the relationships between the *Chsb* genes and different insects, the *Chsb* genes of insects in Lepidoptera, Diptera, Orthoptera, Coleoptera, and Hymenoptera were selected to construct a phylogenetic tree. The results showed that the relationship between *Cnaphalocrocis medinalis* and *H. vitessoides* was the closest in Lepidoptera, having a confidence level of 96% (Figure 3).

### 3.2. Stage-Specific and Tissue-Specific Expression Patterns of HvChsb

*HvChsb* was highly expressed in the growth and development stages of the larvae, which peaked at the L5 larval stage. Subsequently, the expression level of *HvChsb* peaked among all stages on the first day of the pre-pupal stage, and its level then began to decrease rapidly until there was no expression in the pupal stage. Expression was detected in adulthood, though its level was much lower than that at the L1 larval stage (Figure 4).

Expression levels in six different tissues (head, epidermis, foregut, midgut, hindgut, and fat) of the larvae. The results show that the expression of *HvChsb* was not detected in the head and epidermis, and the expression was only present in the foregut, midgut, hindgut, and fat, peaking in the midgut (Figure 5).

The relative expression levels in five regions (head, thorax, abdomen, wings, and feet) of *H. vitessoides* adults showed that only the abdomen, wings, and feet exhibited *HvChsb* expression. Among these regions, *HvChsb* expression peaked in the abdomen, and the expression level in the head was about 350 times that of the control (HD) (Figure 6).

### 3.3. Silencing of HvChsb via RNAi

dsRNA was injected into L3D1 (first day of third instar) larvae, and total RNA was then extracted. The expression level of *HvChsb* after RNAi was determined via RT-qPCR. The results showed that ds*HvChsb* could silence the target gene. The expression of *HvChsb* was lower than that of the control at 12, 24, 36, 48, 72, and 96 h after dsRNA injection. In particular, at 12 h after injection, the interference efficiency was highest, while the *HvChsb* level was lowest (about 60% of the control group) (Figure 7).

### 3.4. Phenotypic Analysis and Survival Assay after RNAi

After successful silencing of *HvChsb*, we compared the control group with groups injected with ds*GFP* and DECP. We found that the individuals injected with ds*HvChsb* exhibited clear lethality and developmental abnormalities (Figure 8A). The survival rate from the larval stage to successful pupation was 43.3%, and this rate was significantly higher in the control than in the experimental group (Figure 8B). The average weights of the experimental and control groups measured at 24 h were significantly different (Figure 8C).

### 3.5. Starvation Treatment and Re-Feeding

L4 larvae of *H. vitessoides* were randomly selected for starvation stress experiments, and their expression levels were determined at 12, 24, 36, 48, 72, and 96 h. The results showed that *HvChsb* expression was significantly inhibited by increasing the starvation time, and it reached its lowest level at 96 h (Figure 9).

L4 larvae of *H. vitessoides* were randomly selected for re-feeding after starvation, and their expression levels were detected at 0.5, 1, 4, and 12 h. The results showed that the expression level of *HvChsb* began to increase when the starved larvae were re-fed at 0.5 h. Upon re-feeding at 12 h, there was no difference between the experimental and control groups (Figure 10).

## 4. Discussion

Previous studies showed that chitin is an important molecule for insects, which is key to forming insect epidermis and PM. Chitin synthesis in insects involves many complex steps, in which CHS is a key component [4]. The study of chitin synthase has a long history, and the genes encoding it were initially simply divided into the genes *Chs1* and *Chs2*; in 2005, researchers instead named these two genes chitin synthase A (*Chsa*) and chitin synthase B (*Chsb*) due to differences in function and specificity [26,28]. Regarding differences between these two genes, the *Chsa* gene is mainly involved in chitin synthesis in insect cuticles, while the *Chsb* gene is expressed in insect midgut PM, which catalyzes the formation of midgut PM [9,10,29,30]. In this study, a *Chsb* gene (*HvChsb*) was obtained and successfully identified from the existing transcriptome of *H. vitessoides* (Figure 1). Homology analysis showed that the amino acid sequence encoded by *HvChsb* had high similarity to *Chsb* of Lepidoptera. The homology was 73.37% with *CmChsb* of *C. medinalis* and 69% with that of other insects. Meanwhile, the homology of *Chsb* with the sequences in other Coleoptera *T. castaneum*, Orthoptera *L. migratoria*, and Diptera *B. dorsalis* was less than 50%. This result indicates that the *Chsb* of insects differs markedly among different orders. The phylogenetic tree constructed using the amino acid sequences showed that *HvChsb* could be divided into two categories. *HvChsb* had the closest relationship with *C. medinalis*, though it also had low homology with *Chsb* of Diptera and Hymenoptera (Figure 3). *Chsb* is particularly expressed in lepidopterans, suggesting that it is involved in growth and development.

Insects of many species differ in terms of the duration of their development, and their expression patterns with age also vary. For example, in the metamorphosis of *Drosophila melanogaster* and *Tribolium castaneum*, the relative expression of the *Chsb* gene differs significantly. In a study on *D. melanogaster*, *DmChsb* was found to be expressed at all developmental stages, albeit peaking during the pre-pupal stage [7]. Meanwhile, *TcChsb* expression in the pre-pupal stage was significantly lower than in the adult stage in *T. castaneum* [8]. In this study, analysis of the expression during *H. vitessoides* development showed that *HvChsb* was highly expressed in the larval stage, which is also consistent with the feeding behavior of this species. Specifically, this species exhibits aggregate feeding during the L1–L2 instar larval stage, and its limited ability to disperse leads to it only feeding on the leaves found around the hatching site. It was reported that the feeding level began to increase and the feeding range was expanded during the L3–L5 instar larval stage. The food intake was significantly reduced during the pre-pupal period, and no food was eaten during the pupal period [24,31]. Therefore, *H. vitessoides* larvae must eat in large quantities to promote their growth and development. The expression level increased continuously in the larval stage and decreased significantly in the pre-pupal stage; expression was not observed in the pupal stage. The development cycle has a certain regularity, indicating that *HvChsb* may also be involved in energy metabolism. The growth and feeding of *H. vitessoides* require a large amount of energy supply; thus, *HvChsb* also shows a certain regularity. The expression level of *HvChsb* was higher in the larval stage, which may require significant energy for feeding behavior; thus, the expression level of *HvChsb* was also higher in this period (Figure 4). The expression level of *HvChsb* is not high in the pupal stage and adult stage. In some cases, at the pupal stage, it is almost undetectable. It is speculated that *H. vitessoides* pupae do not need to eat and exist in a dormant state. Adults of *H. vitessoides* do not need to eat the leaves of *A. sinensis*, supplement nutrition with nectar, and then complete the finishing work [24,32]. In this study, the expression level of *HvChsb* reached its highest on the first day of the pre-pupal stage, a finding that is similar to some other researchers’ results. As the change in gene expression has a great relationship with its function, the specific function of this result can be further studied and explored. The change in gene expression is also closely related to its function. *HvChsb* expression in *H.vitessoides* peaked on the first day of the pre-pupal stage, though the specific function associated with this requires further study. However, the experimental results for *Ostrinia furnacalis* differ within the same lepidopteran order. Specifically, the expression of *OfChsb* on the last day of the L5 instar stage and expression at the pre-pupal stage are relatively consistent, as well as being lower than at other stages [21]. The above results also show that the expression of the *Chsb* gene differs significantly along development in different insects. We speculate that these differences are probably due to different insects having different development periods, as well as differences in the design conditions of each experiment, such as insect breeding conditions and the time interval for collecting samples.

In this study, *HvChsb* was expressed in the foregut, midgut, hindgut, and fat at the larval stage, but especially in the midgut (Figure 5). This result is consistent with the reported expression patterns of *CmChsb* (*Cnaphalocrocis medinalis*), *BdChsb* (*Bactrocera dorsalis*), and *BmChsb* (*Bombyx mori*) in larval tissues [15,17,33]. High expression in the midgut is likely to be associated with eating behavior [18,20,21,34]. During the feeding period of *H. vitessoides*, its activity and feeding behavior increased, which increased the synthesis of chitin in the peritrophic midgut membrane; thus, the expression of *HvChsb* in the midgut also increased. However, contrasting findings were made for *AgChsb* (*Anopheles gambiae*), where expression was higher in the foregut than in the midgut [3]. It is speculated that this outcome may occur because these insects belong to different orders and their digestive systems have different structures. In this study, there was almost no expression of the *Chsb* gene in the head and epidermis of *H. vitessoides* larvae. It is speculated that expression of the *Chsb* gene is mainly concentrated in the digestive tract, where its product specifically catalyzes the formation of PM chitin in the midgut [3,35], while the expression is low or absent in other anatomical regions. The expression pattern of *HvChsb* in adult tissues showed that this gene was expressed in the abdomen, feet, and wings at this stage, but particularly in the abdomen (Figure 6). This outcome is also associated with insect feeding behavior, increasing the synthesis of PM chitin in the midgut [16,20,36].

The fact that different insects exhibit different RNAi silencing effects shows that insects vary in terms of their sensitivity to RNAi [37], as revealed via numerous RNAi experiments. In the *H. vitessoides* experiments, RNAi had a high silencing effect [38,39,40]. In this study, 1 μL of dsRNA was injected into the L3 instar larvae at a concentration of 3 μg/μL. Through comparative analysis of the experimental and control groups, the results showed that the effect of RNAi was detected within 12 h of injection of ds*HvChsb*, and its relative expression reached its lowest level at 12 h after this injection. After that, the relative expression level began to increase. This finding indicates that RNAi had the expected effect of inhibiting *Chsb* expression (Figure 7). After injection of ds*HvChsb*, the growth and development of *H. vitessoides* also showed abnormalities (Figure 8A). Specifically, the larvae in the experimental group developed more slowly than those in the control group. The body color of those in the experimental group became yellow, and the larvae were shorter due to their limited feeding. There were also difficulties in pupation and eclosion. After eclosion, the experimental group’s adults (males and females) were also smaller than those of the control group. In the study of *Cnaphalocrocis medinalis*, *Bombyx mori* and *Spodoptera exigua Chsb* was silenced, the insects’ bodies became smaller, the weights were reduced, and pupation and eclosion were difficult [15,17,41]. Dut to RNA interference, *Chsb* is silenced, and eating is blocked. It is impossible to obtain the energy-providing substances needed for growth and development, which affects growth, pupation, and emergence. We speculate that silencing *Chsb* may lead to a decrease in *Chsa* expression, which increases the possibility of abnormal development. [16,42,43]. We compared the full-length sequences of *HvChsa* (MH142084.1) and *HvChsb*. The similarity was 47.58 %, indicating that the functions of the two genes were different. Due to sequence similarity, the possibility of knocking down *HvChsa* cannot be completely ruled out, which may lead to developmental abnormalities. We will explore this situation further in a later study. After injection of ds*HvChsb*, the survival rate of larvae decreased significantly, especially in the larval stage, and the survival rate from larvae to pupation was only 43.33% (Figure 8B). Similar findings were also made upon silencing *Chsb* genes in other insects. For example, upon interfering with the expression of *BmChsb* in *Bombyx mori* at the larval stage, most larvae could not normally molt [15]. In addition, after silencing the expression of *LmChsb* in *Locusta migratoria* adults, they were unable to digest and absorb food, eventually dying of starvation, with a mortality rate of 78% [19]. It is speculated that silencing the *Chsb* gene mediated via RNAi destroys the mechanism of chitin synthesis in the midgut PM, and the structure and function of the PM are also destroyed accordingly. This result would, in turn, affect insects’ feeding and food absorption and digestion. In this study, the inability of *H. vitessoides* to eat and its lack of energy supply for growth and development eventually led to mortality. At the same time, upon injection in the control group, these phenomena were not observed. These results further indicate that *HvChsb* plays an essential role in the growth and development of *H. vitessoides*, and suggest that it is a key gene required for the substantial feeding that occurs in *H. vitessoides* larvae.

The *Chsb* gene is closely related to insects’ activity and feeding behaviors. The enzyme it encodes is key to the synthesis of chitin in the midgut PM of insects. Experiments involving the induction of starvation stress showed that this process can stimulate the expression of the *Chsb* gene [19,34,44]. Upon starving *L. migratoria* for 24 and 48 h, it was shown that the expression of *LmChsb* in the experimental group was significantly lower than that in the control group. At the same time, the PM was severely damaged in the experimental group, which impaired eating and digestive function. Moreover, the midgut length of the experimental group was also shorter than that of the control group [19]. Meanwhile, in the current exploratory study, *H. vitessoides* at the L4 instar stage was subjected to starvation stress and re-feeding. At 96 h of starvation, *HvChsb* expression in the experimental group was significantly inhibited from 12 h. Over time, its expression continued to drop, reaching its nadir at 96 h (Figure 9). In a further experiment involving re-feeding after starvation, *H. vitessoides* starved for 48 h was re-fed. It was found that *HvChsb* expression began to increase at 0.5 h after re-feeding. In comparison, there was no significant difference in expression between the experimental and control groups at 12 h (Figure 10). This result is consistent with the experimental results from *L. migratoria*. This result further indicates that the function of the *Chsb* gene is closely related to insect-feeding behavior and the formation of PM chitin in the midgut. Previous studies also showed that *Chsb* catalyzes the synthesis of chitin in insect midgut PM [1,4,45]. Silencing the *Chsb* gene would impede insects’ feeding behavior, obstruct their energy supply, and destroy the functional mechanism of PM chitin in the midgut.

## 5. Conclusions

The *Chsb* gene was successfully obtained from the transcriptome database of *H. vitessoides* and identified as *HvChsb. HvChsb* is highly expressed at the larval stage, with its relative expression peaking on the first day of the pre-pupal stage. The detection of *HvChsb* expression in larvae and adults showed that its relative expression levels peaked in the larval midgut and adult abdomen. Moreover, the results of starvation treatment showed that *HvChsb* was significantly inhibited by increasing the starvation duration. Meanwhile, the results of a re-feeding experiment after 48 h of starvation showed that *HvChsb* expression in the experimental group began to grow at 0.5 h of re-feeding. No significant difference compared to the control group was found at 12 h. Furthermore, the injection of ds*HvChsb* resulted in the silencing of *HvChsb*, with the strongest inhibitory effect occurring at 12 h, and the phenotypic abnormalities occurred during growth and development. At the same time, the survival rate of *H. vitessoides* decreased significantly. In this study, the knowledge of the insect *Chsb* gene was enriched, and it provided a reference for the application of RNAi technology in insect control.

## Figures and Tables

**Figure 1 insects-14-00608-f001:**
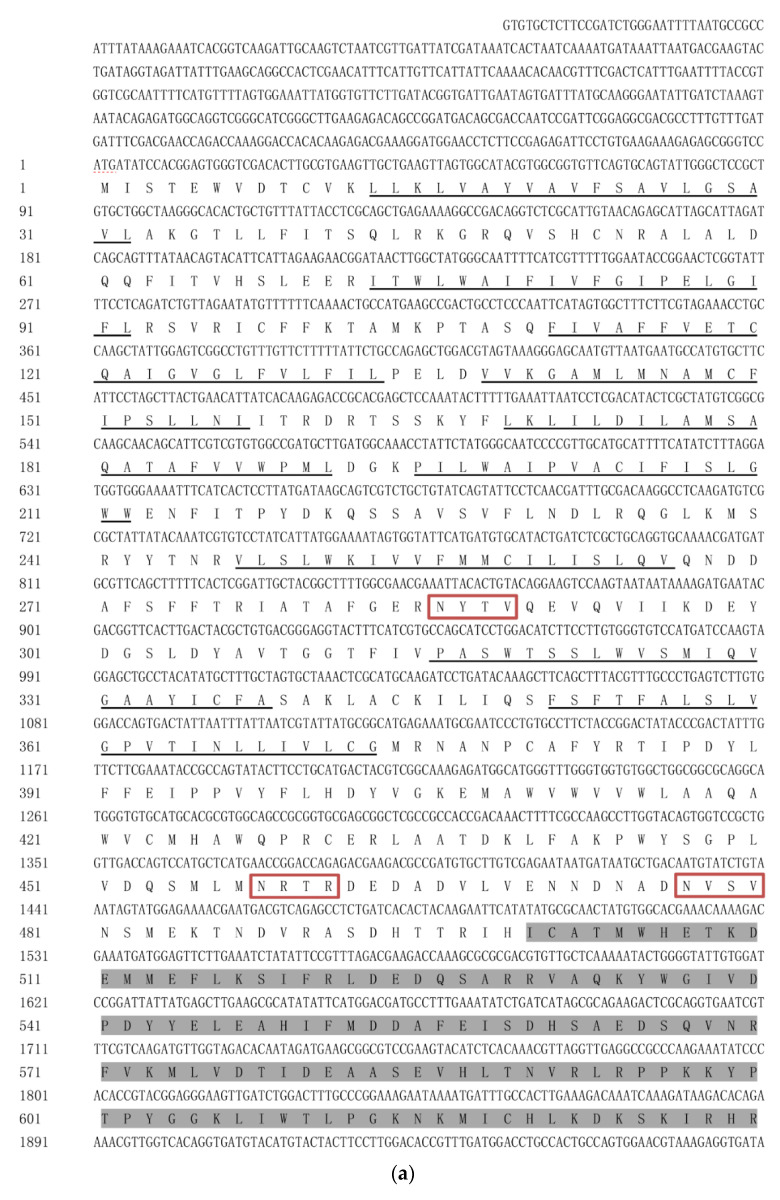
(**a**) The amino acid sequence of *HvChsb* from *H. vitessoides.* (**b**) The amino acid sequence of *HvChsb* from *H. vitessoides.* (**c**) The amino acid sequence of *HvChsb* from *H. vitessoides.* The start codon and the termination codon are marked with a red dotted line, conserved regions are marked with gray shading, the 16 transmembrane domains are marked with black underlines, and the 4 potential N-glycosylation sites are surrounded by red boxes.

**Figure 2 insects-14-00608-f002:**
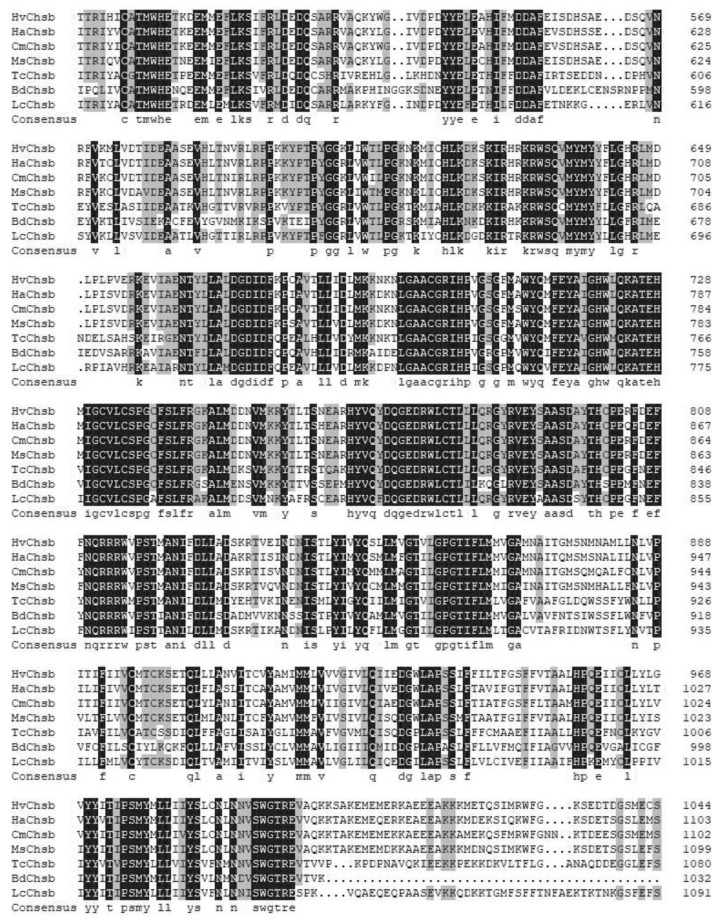
Sequence alignment of *HvChsb* with insect homologs. The amino acid residues that are identical in all sequences are marked with dark shading, whereas light shading indicates that at least 75% amino acids are identical in all sequences. The aligned sequences are the predicted amino acid sequences of Chsbs from *H. vitessoides* (*HvChsb* GenBank accession number ON783456), *Helicoverpa armigera* (*HaChsb* AKZ08595.1), *Cnaphalocrocis medinalis* (*CmChsb* AJG44539.1), *Mythimna separata* (*MsChsb* ASF79498.1), *Tribolium castaneum* (*TcChsb* AAQ55061.1), *Bactrocera dorsalis* (*BdChsb* KC354694.1), and *Locusta migratoria* (*LmChsb* JQ901491.1).

**Figure 3 insects-14-00608-f003:**
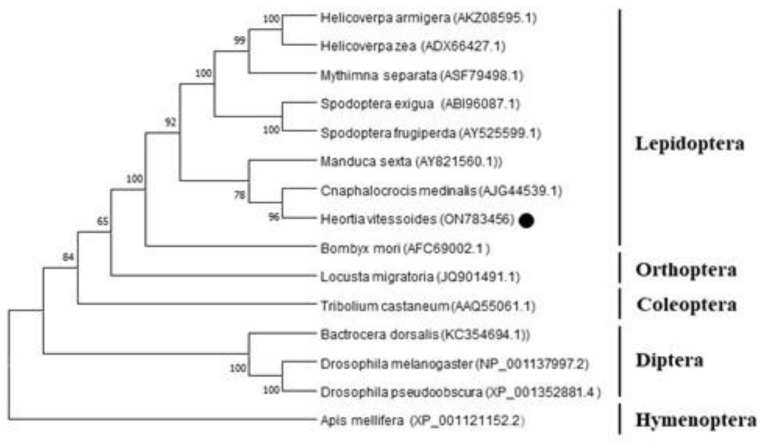
Phylogenetic analysis of *HvChsb*. The predicted amino acid sequences of *HvChsb* together with 14 selected *Chsb* members were aligned, and a phylogenetic tree was constructed using MEGAX. GenBank accession numbers are as follows: *HaChsb*, *H. armigera* (AKZ08595.1); *HzChsb*, *H. zae* (ADX66427.1); *MsChsb*, *M. separata* (ASF79498.1); *SeChsb*, *S. exigua* (ABI96087.1); *SfChsb*, *S. frugiperda* (AY525599.1); *MsChsb, M. sexta* (AY821560.1); *CmChsb*, *C. medinalis* (AJG44539.1); *BmChsb*, *B. mori* (AFC69002.1); *LmChsb*, *L. migratoria* (JQ901491.1); *TcChsb*, *T. castaneum* (AAQ55061.1); *BdChsb*, *B. dorsalis* (KC354694.1); *DmChsb*, *D. melanogaster* (NP_001137997.2); *DpChsb*, *D. pseudoobscura* (XP_001352881.4); and *AmChsb*, *A. mellifera* (XP_001121152.2).

**Figure 4 insects-14-00608-f004:**
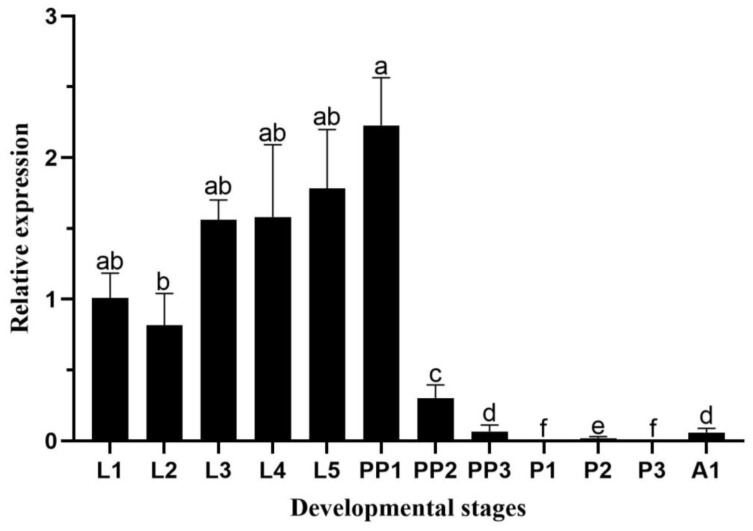
Relative expression levels of *HvChsb* at different stages: L1–L5, first- to fifth-instar larvae; PP1–PP3, 1- to-4-day-old pre-pupae; P1–P3, 1-to-3-day-old pupae; A1, 1-day-old adults. Error bars represent mean ± standard error of three biological replicates. Different letters above error bars indicate significant differences (*p* < 0.05) based on one-way ANOVA and Tukey’s test.

**Figure 5 insects-14-00608-f005:**
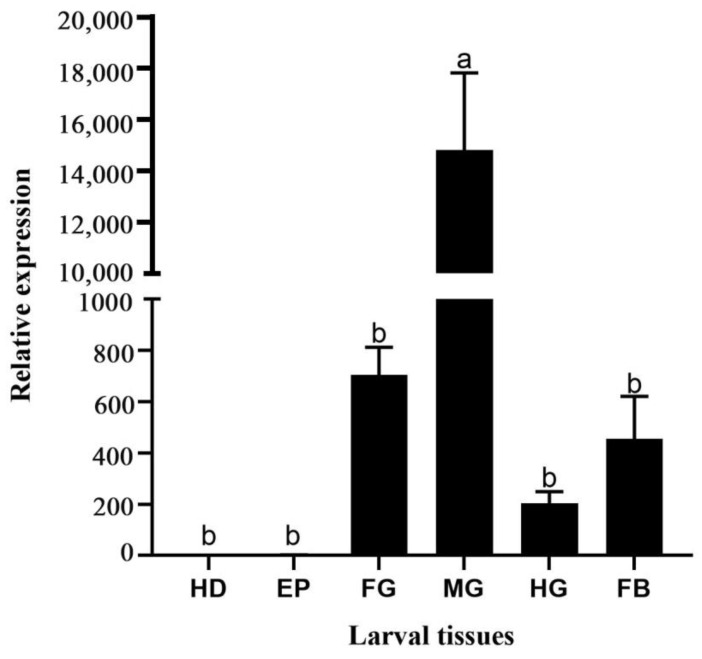
Relative expression levels of *HvChsb* in different larval tissues (tissue anatomy for the fifth-instar larvae). Relative expression in larval tissues: HD, head; EP, epidermis; FG, foregut; MG, midgut; HG, hindgut; and FB, fat body. Error bars represent mean ± standard error of three biological replicates. Different letters above error bars indicate significant differences (*p* < 0.05), which were based on one-way analysis of variance (ANOVA) and Tukey’s test.

**Figure 6 insects-14-00608-f006:**
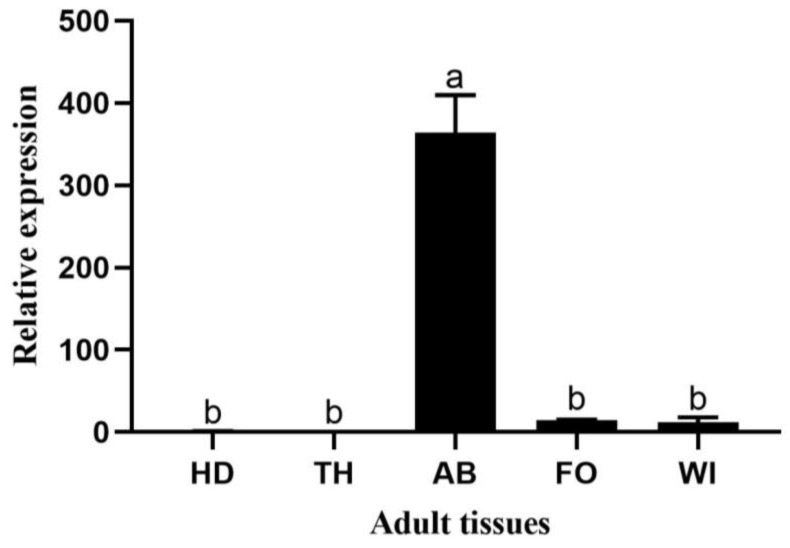
Relative expression in adult tissues: HD, head; TH, thorax; AB, abdomen; FO, foot; and WI, wing. Error bars represent mean ± standard error of three biological replicates. Different letters above error bars indicate significant differences (*p* < 0.05), which were based on one-way analysis of variance (ANOVA) and Tukey’s test.

**Figure 7 insects-14-00608-f007:**
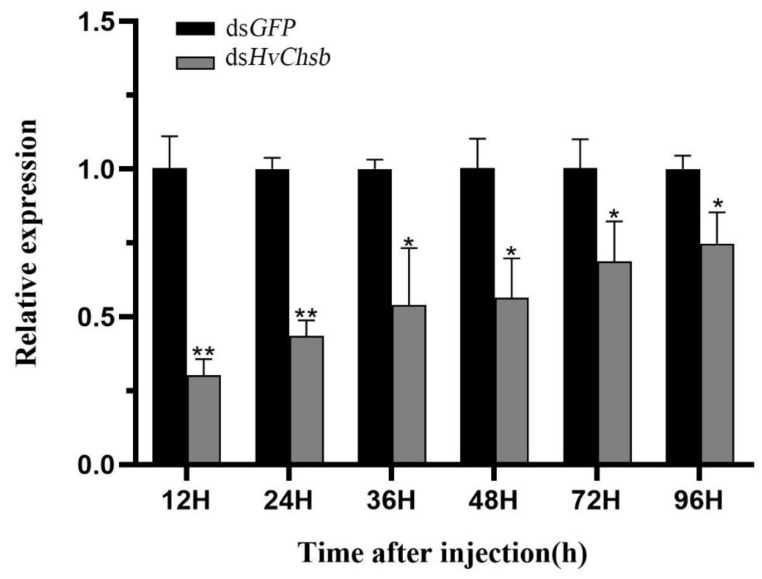
Changes in mRNA level after treatment with specific RNA interference. Relative transcript levels of *HvChsb* in L3D1 larvae after injection with ds*HvChsb* at a concentration of 3.0 µg/µL for 12, 24, 36, 48, 72, and 96 h. The sample size was 120 larvae, which were divided into three biological replicates. Error bars represent mean ± standard error of three biological replicates. * *p* < 0.05, ** *p* < 0.01. Analysis was performed via one-way analysis of variance (ANOVA), followed by Student’s *t*-test.

**Figure 8 insects-14-00608-f008:**
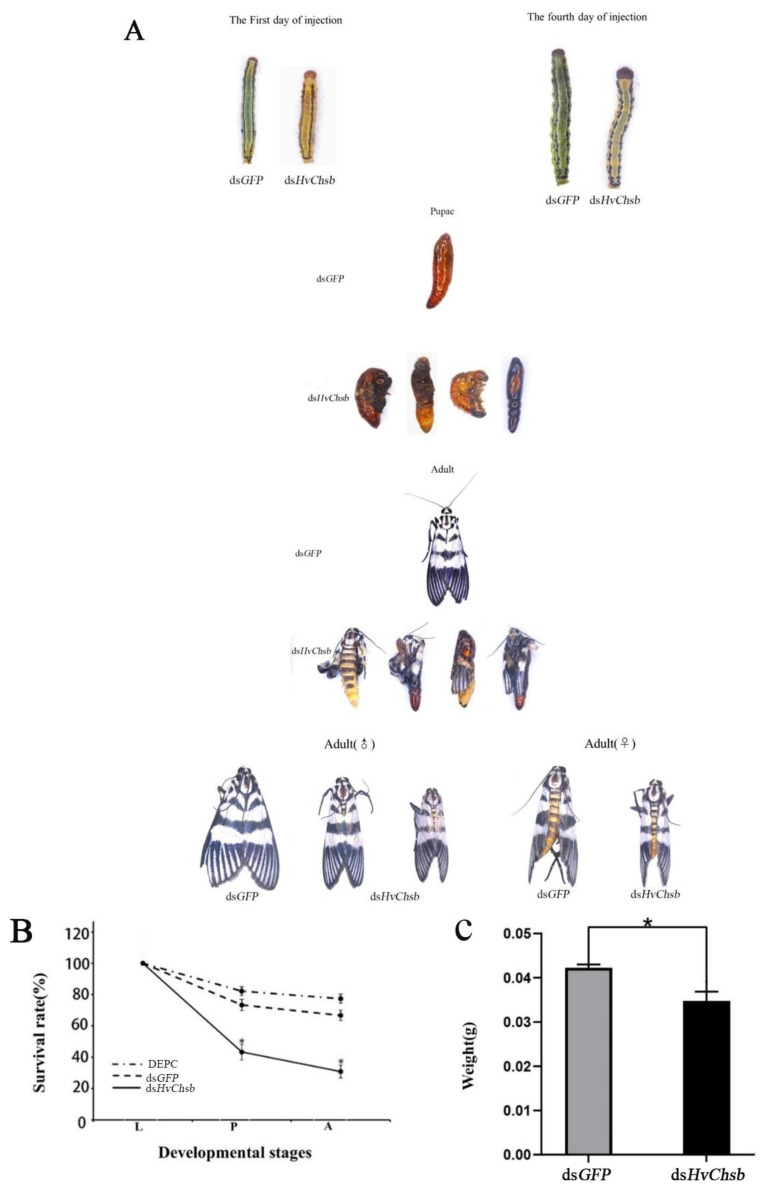
Effects of *HvChsb* RNAi on larval-to-pupal and pupal-to-adult molting. (**A**) Data on developmental abnormalities or lethality due to the RNAi treatment of *HvChsb* are shown as the mean ± standard error of three biological repeats. (**B**) Effects of *HvChsb* RNAi on larval-to-pupal and pupal-to-adult transition rates. Rates of insect survival from fifth-instar larval stage to adulthood after ds*HvChsb* injection (* *p* < 0.05, Kaplan–Meier survival analysis with log-rank test). Data are the mean ± standard error of three biological repeats. (**C**) Larval weight at 24 h after ds*HvChsb* and ds*GFP* injections. These data were recorded separately based on a sample size of 120 larvae. Error bars represent mean ± standard error of three biological replicates.* *p* < 0.05, based on Student’s *t*-test.

**Figure 9 insects-14-00608-f009:**
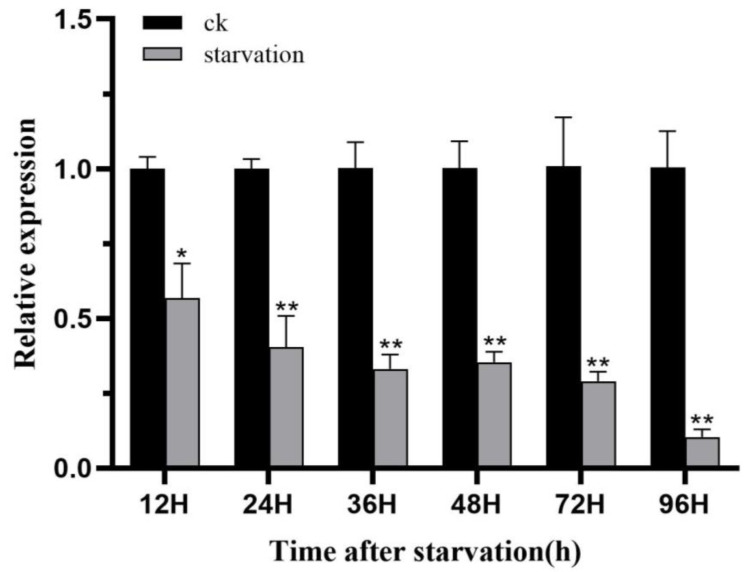
Expression profiles of *HvChsb* after 96 h of starvation. Expression levels at 12, 24, 36, 48, 72, and 96 h after starvation were normalized compared to those at 12, 24, 36, 48, 72, and 96 h after feeding (control). * *p* < 0.05; ** *p* < 0.01 (*t*-test). Data are the mean ± standard error of three biological repeats.

**Figure 10 insects-14-00608-f010:**
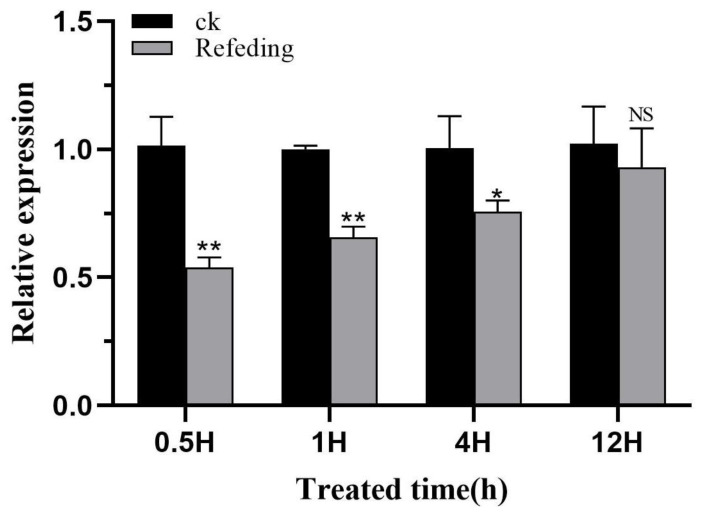
The expression profile of *HvChsb* upon re-feeding. After 48 h of starvation, the insects were re-fed. The expression levels at 0.5, 1, 4, and 12 h were standardized with the expression levels of the control group. * *p* < 0.05; ** *p* < 0.01 (*t*-test). No significant difference between the two groups was represented by NS. Data are mean ± standard error of three biological replicates.

**Table 1 insects-14-00608-t001:** Primers used for RT-qPCR and synthesis of ds*Chsb* and ds*GFP.*

Primer Name	Sequence(5′–3′)	Tm	Product Length (bp)
*HvChsb*-F	CCGCCCAAGAAATATCCCACAC	59.54	344
*HvChsb*-R	GCCATAAAACCAGAGCCAACCG	59.54	
*β*-actin-F	GTGTTCCCCTCTATCGTGG	57.32	119
*β*-actin-R	TGTCGTCCCAGTTGGTGAT	55.11	
T7+ds*HvChsb*-F	taatacgactcactatagggCGTTTGCCCT GAGTCTTG	75.9	514
T7+ds*HvChsb*-R	taatacgactcactatagggTTTCGTCTTT TGTTTCGT	71.1	
T7+ds*GFP*-F T7+ds*GFP*-R	taatacgactcactatagggCAGTTCTTGT TGAATTAGATG taatacgactcactatagggTTTGGTTTGT CTCCCATGATG	71.5 75.5	400

## Data Availability

The data presented in this study are available upon request from the corresponding author.

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
