# Peer review of "Characterization of Chitin Synthase B Gene (HvChsb) and the Effects on Feeding Behavior in Heortia vitessoides Moore"

_insects, 2023, doi:10.3390/insects14070608_

Round 1

Reviewer 1 Report

The manuscript titled "Characterization of Chitin Synthase B Gene (HvChsb) and the Effects on Feeding Behavior in Heortia vitessoides Moore" shows how chitin plays and important role in growth and development of this Crambidae insect.

A minor comment on the manuscript, do add Tm and amplicon size for both RT-qPCR and dsChsb, dsGFP primers in Table 1 and also the efficiency of the dsChsb and dsGFP genes.

Author Response

Dear Reviewer,

Thank you very much for your comments on my manuscript.The revisions of the manuscript are as follows:

Add the content in Table 1, including Tm and amplicon size.

Reviewer 2 Report

The submitted manuscript focusses on the function of the Chsb gene encoding a chitin synthase expressed in the midgut of the moth Heortia vitessoides, a major pest of the agarwood tree Aquilaria sinensis.  The authors have cloned the cDNA and determined the expression profile in different developmental stages and tissues. They further showed that Chsb expression is depending on the nutritional state. RNAi to silence Chsb expression results in reduced survival rates and aberrant development during the larval-to-pupal molt. The manuscript is well written, but the study is largely confirmative as it recapitulates many experiments that have been done before with similar outcomes in other lepidopteran species such as Bombyx mori or Manduca sexta, or other insects amenable to RNAi. However, the finding that RNAi to silence Chsb affects metamorphosis is per se interesting but it needs some control to exclude off-target effects.

Specific comments:

Fig. 2. The amino acid alignment is fuzzy and difficult to read.

Fig. 3. Bootstrap values below 80% do not really support phylogenetic trees well. Does a ML tree give the same results within the lepidopteran group as the NJ tree? Did you use the complete amino acid sequences for calculating the tree? Please indicate.

Fig. 7. As frequently observed in lepidopteran species, RNAi efficiency is poor after three days of injection; however, the effects on development are significant. Did the authors check for off-target effects of closely related genes such as Chsa, which is also known to result in a larval-to-pupal arrest once silenced? This control is important for the main conclusion.

Author Response

Dear Reviewer,

Thank you very much for your comments on my manuscript. The revisions of the manuscript are as follows:

(1) The references were reviewed, and the references were related to the content of the manuscript.

(2) Fig 2, The picture has been modified and replaced, which has been reflected in the revised manuscript.

(3) Fig 3, The phylogenetic tree is to better summarize the genetic relationship between various organisms. It is near and far, will help people understand the historical process of biological evolution. Other articles will also show the complete relationship between distant and close relatives:

  1. Jie Chen, Bin Tang, Hongxin Chen, Qiong Yao, Xiaofeng Huang, Jing Chen, Daowei Zhang, Wenqing Zhang*. 2010.Different Functions of the Insect Soluble and MembraneBound Trehalase Genes in Chitin Biosynthesis Revealed by RNA Interference. PLoS ONE, Volume 5, Issue 4, e10133.
  2. Zhixing Li, Zihao Lyu, Qingya Ye, Jie Cheng, Chunyan Wang and Tong Lin *. 2020. Cloning, Expression Analysis, 20-Hydroxyecdysone Induction, and RNA Interference Study of Autophagy-Related Gene 8 from Heortia vitessoides Moore. Insects. 11, 245.
  3. Chitvan Khajuria, Lawrent L. Buschman, Ming-Shun Chen, Subbaratnam Muthukrishnan, Kun Yan Zhu*. 2010. A gut-specific chitinase gene essential for regulation of chitin content of peritrophic matrix and growth of Ostrinia nubilalis larvae. Insect Biochemistry and Molecular Biology. 40, 621-629.

In this paper, the results of ML tree and NJ tree are the same in Lepidoptera. The complete amino acid sequence was used to construct the phylogenetic tree. Bootstrap analysis results of 1000 replicates are shown.

(4) Fig 7, In this paper, we only did Chsb gene research, not Chsa gene research. In the study of Chsb gene, the specific detection of the designed primers has high specificity. About Chsa gene, it is in mRNA expression, Chsa specific expression in the trachea and body wall formation. Chsa is mainly involved in the synthesis of epidermis and trachea in different periods of insect growth. Effective silencing of Chsa gene. Insects have molting difficulties and molting time delays, tracheal malformations and developmental metamorphosis are blocked to death.

Round 2

Reviewer 2 Report

Dear authors,

Thank you for efforts in improving your manuscript.  Please find my comments to your response letter.

Fig 2. The picture has been modified and replaced, which has been reflected in the revised manuscript.

Response: much better now, thank you.

Fig 3, The phylogenetic tree is to better summarize the genetic relationship between various organisms. It is near and far, will help people understand the historical process of biological evolution. Other articles will also show the complete relationship between distant and close relatives

Jie Chen, Bin Tang, Hongxin Chen, Qiong Yao, Xiaofeng Huang, Jing Chen, Daowei Zhang, Wenqing Zhang*. 2010.Different Functions of the Insect Soluble and MembraneBound Trehalase Genes in Chitin Biosynthesis Revealed by RNA Interference. PLoS ONE, Volume 5, Issue 4, e10133.

Zhixing Li, Zihao Lyu, Qingya Ye, Jie Cheng, Chunyan Wang and Tong Lin *. 2020. Cloning, Expression Analysis, 20-Hydroxyecdysone Induction, and RNA Interference Study of Autophagy-Related Gene 8 from Heortia vitessoides Moore. Insects. 11, 245.

Chitvan Khajuria, Lawrent L. Buschman, Ming-Shun Chen, Subbaratnam Muthukrishnan, Kun Yan Zhu*. 2010. A gut-specific chitinase gene essential for regulation of chitin content of peritrophic matrix and growth of Ostrinia nubilalis larvae. Insect Biochemistry and Molecular Biology. 40, 621-629.

 In this paper, the results of ML tree and NJ tree are the same in Lepidoptera. The complete amino acid sequence was used to construct the phylogenetic tree. Bootstrap analysis results of 1000 replicates are shown.

Response: Well if your intention is to recapitulate well-established phylogenetic relations within lepidopterans, I wonder why you show this tree at all, as it contains no new information.

(4) Fig 7, In this paper, we only did Chsb gene research, not Chsa gene research. In the study of Chsb gene, the specific detection of the designed primers has high specificity. About Chsa gene, it is in mRNA expression, Chsa specific expression in the trachea and body wall formation. Chsa is mainly involved in the synthesis of epidermis and trachea in different periods of insect growth. Effective silencing of Chsa gene. Insects have molting difficulties and molting time delays, tracheal malformations and developmental metamorphosis are blocked to death.

Response: The new finding in your work is that RNAi for Chsb results in aberrant metamorphosis (which is also observed when chsa is silenced). My concern is that this RNAi phenotype could be due to cross-silencing of chsa and that it does not result from silencing chsb. Therefore, I ask for a control that shows that chsa expression remains unaffected. This is an easy qPCR control experiment, which could be accpmoplished within one day if you still have the cDNAs.

Author Response

Dear Reviewer,

Thank you very much for your comments on my manuscript. The revisions of the manuscript are as follows:

1.The literature in the manuscript was adjusted.

2.Phylogenetic tree:The purpose of showing this phylogenetic tree is to illustrate the close genetic relationship between HvChsb and Lepidoptera, as well as between different orders, so as to facilitate readers to know.

3.After reviewing the literature of chitin synthase A gene (Chsa) and chitin synthase B gene (Chsb), it is more interested in Chsb, so the gene is studied. We also screened the full sequence of Chsa (HvChsa). The amino acid sequences of Chsa and Chsb were compared, and the similarity was 47.58 %, indicating that there were some differences in function between the two genes. In this study, HvChsb was effectively silenced, the feeding of larvae was blocked, and the larvae became yellow and shortened, the weight was reduced. The adult body size was significantly smaller, and pupation and eclosion were difficult. This appeared in the study of Cnaphalocrocis medinalis, Bombyx mori and Spodoptera exigua. Due to RNA interference, Chsb is silenced, eating is blocked, and growth and development energy is missing. It is speculated that Chsb silencing leads to the decrease of Chsa expression, which leads to the difficulty of pupation and eclosion of insects.Amino acid sequence alignment of Chsa and Chsb

(1).Zhang Z, Xia L, Du J, Li S, Zhao F. 2021. Cloning, characterization, and RNAi effect of the chitin synthase B gene in Cnaphalocrocis medinalis. Journal of Asia-Pacific Entomology, 24, 486–492.

(2).Zhuo W, Chu F, Kong L, Tao H, Sima Y, Xu S. 2014. Chitin synthase B: A midgut-specific gene induced by insect hormones and involved in food intake in Bombyx mori larvae. Archives of Insect Biochemistry and Physiology, 85, 36–47.

(3).Hyun Soo Kim, Soyoung Noh, Youngjin Park.2017. Enhancement of Bacillus thuringiensis Cry1Ac and Cry1Ca toxicity against Spodoptera exigua (Hübner) by suppression of a chitin synthase B gene in midgut. Journal of Asia-Pacic Entomology, 20, 199-205.

In the discussion part of the manuscript, the discussion of pupation and eclosion difficulties was modified.

We are deeply grateful to the reviewer for their guidance on the manuscript.

Thank you and best regards.

Round 3

Reviewer 2 Report

The authors have sufficiently addressed my comments. The addition in the discussion is fine, as it appears to be difficult to do the off-traget control. Maybe you better write that you cannot completely rule out the possibility of knocking down HvChsa due to sequence similarity, which might cause an aberrant molting phenotype.

Author Response

Dear Reviewer,

Thank you very much for your comments on my manuscript. 

Based on your comments, we have added to the discussion section of the manuscript.

Thank you and best regards.
